# Numerical Simulation on In-plane Deformation Characteristics of Lightweight Aluminum Honeycomb under Direct and Indirect Explosion

**DOI:** 10.3390/ma12142222

**Published:** 2019-07-10

**Authors:** Xiangcheng Li, Yuliang Lin, Fangyun Lu

**Affiliations:** College of Liberal Arts and Science, National University of Defense Technology, Changsha 410073, China

**Keywords:** aluminum honeycomb, deformation modes, shock wave, counter-intuitive behavior, energy distribution

## Abstract

Lightweight aluminum honeycomb is a buffering and energy-absorbed structure against dynamic impact and explosion. Direct and indirect explosions with different equivalent explosive masses are applied to investigate the in-plane deformation characteristics and energy-absorbing distribution of aluminum honeycombs. Two finite element models of honeycombs, i.e., rigid plate-honeycomb-rigid plate (RP-H-RP) and honeycomb-rigid plate (H-RP) are created. The models indicate that there are three deformation modes in the *X1* direction for the RP-H-RP, which are the overall response mode at low equivalent explosive masses, transitional response mode at medium equivalent explosive masses, and local response mode at large equivalent explosive masses, respectively. Meanwhile, the honeycombs exhibit two deformation modes in the *X2* direction, i.e., the expansion mode at low equivalent explosive masses and local inner concave mode at large equivalent explosive masses, respectively. Interestingly, a counter-intuitive phenomenon is observed on the loaded boundary of the H-RP. Besides, the energy distribution and buffering capacity of different parts on the honeycomb models are discussed. In a unit cell, most of the energy is absorbed by the edges with an edge thickness of 0.04 mm while little energy is absorbed by the other bilateral edges. For the buffering capacity, the honeycomb in the *X1* direction behaves better than that in the *X2* direction.

## 1. Introduction

Aluminum honeycomb is a promising lightweight structure—a typical bionic structure [1,2,3]. Due to the homogeneous deformation mechanism [4], high strength to weight ratio [5], and large specific energy absorption [6,7], aluminum honeycombs have been frequently applied in many important engineering fields in recent years, such as aircraft, aerospace, transportation, building, and sporting equipment, which are involved in the dynamic impact or indirect explosive load. Hexagonal aluminum honeycomb, as one of the most widely used honeycombs, is orthotropic, i.e., one out-of-plane direction and two in-plane directions, respectively. Dynamic impact and indirect explosive load have been studied widely, while the compressive response of honeycomb structures subjected to direct explosive load is rarely researched. Hence, there is a pressing need to better understand the deformation behaviors of honeycombs subjected to direct explosion.

Numerical simulation is an effective way to study the deformation properties of honeycombs because it is difficult to observe the deformation modes of honeycombs experimentally. Therefore, there are some researchers who have conducted their work by numerical simulations [8,9,10,11,12,13]. For example, Ashab [8] investigated the mechanical behavior of aluminum hexagonal honeycombs subjected to dynamic indentation and compression loads only using ANSYS/LS-DYNA. Qiao [9] explored the in-plane uniaxial collapse response of a hierarchical honeycomb only by finite element simulations. Besides, there are some analogous deformation modes in references [8,9] to that studied in this work. Ruan et al. [10] observed three modes in one of the in-plane directions, X mode at low velocity, V mode at moderate velocity, and I mode at high velocity. Besides, V mode was also observed at low velocity and I mode at high velocity in the other direction. Hu et al. [11] obtained the same conclusion. Moreover, the cell-wall angle, the crushing velocity, and the honeycomb’s relative density could determine the in-plane dynamic properties of hexagonal honeycomb at different cell-wall angles. Wang [14,15] studied the combined effects with a dynamic inclined load. Oblique loads [16] in both in-plane and out-of-plane directions produced deformation modes from X, V, I modes to Half-X mode, Half-X following I mode, V mode, and I mode. Sun et al. [17] concluded that the impacting velocities were related to the stress when all configuration parameters were kept constant. Numerical models could adequately approximate the study of the in-plane deformation properties of honeycombs.

In engineering applications, such as explosion-proof tank, weapon protection engineering, it is possible that honeycomb is subjected to explosive loading. Li et al. [18] investigated the blast resistance of a square sandwich including hexagonal aluminum honeycomb cores, with different cell side lengths of the core. They concluded that failure modes of face sheets were related to the scaled distance of the explosive and the deformation modes of honeycomb core included full densified, progressive buckling, shear deformation, and fragments. Zhu et al. [19] studied the dynamic properties of a sandwich structure with a honeycomb core by air blast experiments. They thought deformation modes were mainly related to face sheet thickness and honeycomb core density. Li et al. [20] investigated the pressure distribution and the structural response of a sandwich with graded honeycomb cores. Three deformation modes were classified: large inelastic bending deformation without a significantly localized core compression, large inelastic bending deformation with significantly localized core compression, and large inelastic bending deformation with localized penetration. Jin et al. [21] proposed an innovative sandwich structure with an auxetic re-entrant cell honeycomb core and investigated its blast resistance by simulation. They showed that both the graded honeycomb cores and the cross-arranged honeycomb cores were better than the ungraded honeycomb cores and regular-arranged cores in the resistance ability. More work with different configurations or various combined structures were also studied by explosive loading [22,23,24,25]. A common feature of these studies is the indirect interaction between explosive loading and honeycomb structures.

According to the literature above, deformation properties of honeycomb under dynamic impact or indirect explosive loading have mostly been analyzed, but it is rare to study in-plane deformation characteristics of aluminum honeycomb structure core impacted by direct blast load. There are two typical applications that led to this work. One is as a passive explosive wave energy sensor, which has the characteristics of a low-cost, no electricity measurement and shock wave energy measurement under bad environment, while the other is as a shock absorber in the spaceflight separator, which can effectively reduce the shock to protect the electronic equipment on the test platform. However, deformation characteristics and energy distribution of honeycomb structures subjected to direct explosive loads are unclear.

This work focuses on the direct interaction between the honeycomb structure and spherical blast wave. The layout of the rest of the paper is arranged as follows. In Section 2, a hexagonal honeycomb structure is proposed with a cell length of 2 mm and 0.04 mm thickness with four edges and 0.08 mm with two edges. Besides, double finite element models in Section 3, including honeycomb-rigid plate (H-RP) and rigid plate-honeycomb-rigid plate (RP-H-RP) in the two in-plane directions with a 50 cm fixed explosive height, are created. Then, in Section 4, deformation modes of the models are described, and more details are given about the deformation properties and mechanism. In particular, the energy distribution of different parts and contact stress comparisons during the whole process are discussed and analyzed in Section 5, followed by a conclusion in Section 6. It is noted that the study is only a preliminary numerical example, which provides a new idea to explore the in-plane deformation regularity of honeycombs under direct explosion by numerical simulation. This work enables the improvement of the energy absorption and buffering capacity of honeycomb used in an explosive environment.

## 2. Honeycomb Core and Loading Setup

### 2.1. Honeycomb Core

An orthotropic hexagonal honeycomb structure (Figure 1a) is constructed by some regular hexagonal cells (Figure 1b) using the stretch forming method. Briefly, the direction along the stretch forming of the honeycomb structure is the *X1* direction. The vertical direction of *X1* in the in-plane is referred to as the *X2* direction, and *X3* is along the out-of-plane direction. The in-plane dimension of the aluminum honeycomb sample (*X1*-*X2*) is 47.6 mm × 46.0 mm with a height of 4 mm, based on an experimental honeycomb sample, and the number of repeated unit cells in the *X1* and *X2* directions is 14 and 15, respectively. The sketch of a unit cell is shown in Figure 1b. The edge length of the unit is 2 mm and the angle between each adjacent edge is 120°. Due to the processing method, the cell wall thickness in the *X1* direction, which is 0.08 mm, is twice that of the other directions.

### 2.2. Loading Setup

The present work creates two types of models under air blast, which are the honeycomb-rigid panel (H-RP) and the rigid panel-honeycomb-rigid (RP-H-RP), respectively, as sketched in Figure 2. The H-RP is used to analyze the mechanical and energy-absorbing properties of honeycomb structures subjected to the direct shock wave. For comparison, RP-H-RP is subjected to the indirect shock wave, where deformation of the rigid plate is ignored. In both models, the bottom rigid plate is fixed. Besides, a spherical explosive wave is realized by an air explosion. The honeycomb is loaded in both the *X1* direction and the *X2* direction, respectively, for each model. Besides, the height of the burst in this work is fixed as 50 cm, which describes the distance between the explosive and the ceiling of the model.

## 3. Numerical Simulation

### 3.1. Finite Element Models

Numerical simulation is conducted by ANSYS/LS-DYNA (Version 971), a popular software to simulate the dynamic impact or detonation. Based on two types of models in Section 2.2, related finite element in-plane models are created, as shown in Figure 3. Three parts are included in the H-RP (Figure 3a), in which Part 1 is the cell’s double edges with 0.08 mm thickness (marked as DE), Part 2 is the cell’s single edges with 0.04 mm thickness (marked as SE), and Part 3 is the bottom rigid plate which is constrained in all directions. The ceiling of the honeycomb is a loaded surface. In addition, there are four parts in the RP-H-RP (Figure 3b). In this model, the former parts are the same as those in the H-PR, while Part 4 is the ceiling rigid plate with 0.08 mm thickness. The plate, as the loaded surface in the RP-H-RP, is only translated in the load direction. Besides, the size of all the grids is set as 0.125 mm to enable steady calculation. Both models are created from the loading directions of *X1* and *X2*, respectively, to research the effect of different loading surface shapes on the deformation characteristics.

### 3.2. Material Properties and Control Setup

Here, the keyword LOAD_BLAST is used to define an air blast function for the application of pressure loads to explosives in conventional weapons, in which trinitrotoluene (TNT) is used as the explosive to simulate the detonation in the keyword file. Related parameters of TNT are taken from reference [26]. Besides, MAT_PLASTIC_KINEMATIC is a material model in the ANSYS/LS-DYNA (Version 971) software to describe the honeycomb structure of the parent material aluminum alloy 3003H18. MAT_RIGID is a rigid material to describe the rigid plate in the bottom/ceiling, where steel material parameters are used to replace parameters of the rigid body. The specific material parameters are shown in Table 1.

In these models, all finite elements are quadrilateral 2D grids, where the thickness of the SE is 0.04 mm, the thickness of the DE is 0.08 mm, and the type of these elements is the Belytschko-Tsay Shell 163 with five through-thickness integration points to provide simulation accuracy. All nodes limit the freedom of translation in the *X3* direction. It is noted that additional components should be installed to prevent the out-of-plane deformation in actual applications. As for contacts, the automatic double-sided face is employed between parts of the models. Self-contact is used for the bucking contact of the same part. Hourglass energy and energy dissipation are computed and included in the energy balance to make the simulation accuracy. The scale factor for the computed time step is 0.9 and the output time step of energy and force is 5 μs. The shell elements between the different parts of the honeycomb structure are connected by common nodes. The main variable in the work is the equivalent mass of the TNT controlled by LOAD_BLAST. Three typical cases with equivalent explosive masses of 10 g, 20 g, and 40 g are employed in the work.

In addition, the effect of different grid densities is analyzed. As a result, the elements with 0.125 mm are created in the following discussion. Besides, the mesh size also fits well with the numerical simulation results in reference [8]. However, if the mesh is further refined, it will have less and less influence on the deformation result, but it will take more time to calculate the model. Therefore, it could not only guarantee the efficiency of calculation but also make simulation results consistent.

In order to analyze the effect of grid density, three sizes of grids are selected, 0.5 mm, 0.25 mm, and 0.125 mm, respectively. Taking the 10 g TNT explosive as an example, the deformation characteristics of the different mesh sizes at four typical moments are obtained, as shown in Figure 4. At the moment of 1700 μs, the difference in both the overall deformation and the local deformation is obvious. Concretely, the overall deformation increased from 0.26 to 0.32 with the size of the elements from 0.5 mm to 0.125 mm. Besides, the concentrated deformation zone in the local position was more prominent in the model with a mesh size of 0.125 mm. It should be noted that the compressive deformation strain is the ratio of the displacement of the upper panel to the length of the honeycomb compression direction, marked as a symbol, *ε*.

## 4. Deformation Modes

### 4.1. The X1-RP-H-RP

With the increase of equivalent explosive masses (10 g, 20 g, and 40 g), there are three deformation modes in the *X1* direction, the overall response mode in the small equivalent explosive masses case, the transitional response mode in the medium equivalent explosive masses case, and the local response mode in the large equivalent explosive masses case, respectively, as shown in Figure 5. In this figure, the upper figures describe the deformation of the whole honeycomb structure, and the bottom figures record the deformation of the SE in order to observe the deformation characteristics clearly. In the bottom figures, the red lines are used to describe the shape of the local deformation belt. Besides, the subsequent description of the deformation characteristics is also presented in this way. The specific meanings of the three deformation modes are as follows, in which the deformation diagrams on display are selected based on whether the local deformation is clearly visible.

The overall response mode means all parts in the model take part in the process of deformation. In the deformation mode, the honeycomb sample is compressed by the upper panel, but the compact state of the honeycomb is not achieved. In detail, at 550 μs, the shock wave begins to propagate into the honeycomb structure. At the moment of 1000 μs, the compressive strain of the honeycomb structure is up to 0.14. A “C” shape folding interface appears near the impact section. Then, an “X” folding interface (or double “C” folding interface back to back) occurs near the upper plate at 1360 μs and a “C” interface appears near the bottom plate. At the moment, the honeycomb structure is compressed to the maximum deformation which is 0.18. After that, the upper panel starts to rebound with a constant speed of 5 m/s at 1800 μs because of the elastic potential energy stored in the honeycomb structure and the compression strain decreases to 0.15. Besides, an interesting phenomenon occurs that some cells near the central axis keep the regular hexagonal configuration in the whole process, and it seems that the cells never received any load.

In the transitional response mode, the honeycomb sample can be exactly compact. The blast wave reaches the honeycomb structure at 455 μs. At 800 μs, the compression strain of the whole honeycomb structure is 0.25, which has exceeded the maximum deformation in the overall response mode. At this moment, the clear deformation interface in the honeycomb structure is a mixed buckling region of “I” and “C” shape, while the area near the upper panel shows the local compaction state and presents the “I” shape. Then, the shape of “C” becomes plain gradually and becomes the “K” mode at the moment of 1200 μs. Once the folding degree of the cells close to the upper panel at the end of the impact section increased to 0.62 at 1800 μs, a larger span of “K” shape occurs. At last, the upper panel bounces back at 6.7 m/s. In addition, the “regular cells” mentioned in the overall response mode disappear in the transitional response mode.

The local response mode is similar to the high-speed loading of deformation proposed by Hu [8]. At 375 μs, the honeycomb structure begins to be loaded by a shock wave. On the upper panel, a local “I” shape interface appears in the honeycomb structure at 570 μs, while honeycomb cells near the bottom rigid plate are not deformed and then at 795 μs, the “I” shape interface is clearer. The maximum compression strain is 0.88 at 1400 μs. The sample reaches the compact state in the mode. The upper panel rebounds back eventually at 7.4 m/s.

### 4.2. The X2-RP-H-RP

The *X2* direction is perpendicular to the *X1* direction in the in-plane region. In order to compare the deformation difference with the *X1*-RP-H-RP, the explosion load is also applied in the *X2*-RP-H-RP. Two types of deformation modes occur, expansion mode in the small equivalent explosive masses case and local inner concave mode in the large equivalent explosive masses case, respectively, as illustrated in Figure 6.

Taking the 10 g TNT as an example of the small equivalent mass cases, the honeycomb sample is expanded across the whole compression process in the expansion mode, but it fails to reach the compact state. In this model, the time for the explosive wave to reach the loading surface is the same as that in *X1*-RP-H-RP, both of which are 550 μs, because they have the same explosive yield and blast height. At 890 μs (ε = 0.14), there is an obvious lateral expansion by the way of stretching transversely of the SE. The mechanism makes the cell shape change from the convex hexagon to the square or even concave hexagon. At the moment of 1260 μs, the deformation reaches the maximum, i.e., *ε* = 0.18. The impacted region forms “∇“ shape area, a series of “I” interfaces. At last, the overall elasticity properties make the ceiling plate rebound with 5.7 m/s. The case of 20 g is similar to the case of 10 g. Local inner concave mode occurs at large equivalent explosive masses (for example, 40 g TNT in Figure 6b). The explosion shock wave begins to propagate in the honeycomb sample at 375 μs. At the moment of 560 μs, the strain of the honeycomb sample is compressed to be 0.25. An “I” interface appears near the impact region. Then, a “bow” shape forms at 790 μs (ε = 0.50) and an “M” interface becomes contour of honeycomb structure at 1570 μs (ε = 0.88).

### 4.3. H-RP

In-plane deformation characteristics of the honeycomb sandwich structures (rigid plate-honeycomb-rigid plate) are obtained with the above two models. In order to obtain the effect of the direct explosive load on the honeycomb and analyze the influence of the shape of the explosive load surface on the deformation characteristics of honeycomb, H-RP is studied in this work. Taking 10 g *X1*-H-RP and 10 g *X2*-H-RP as examples, as illustrated in Figure 7, the same moments are used to compare with the previous two models. Similarity and unique deformation characteristics in the H-RP are shown as follows.

The similarity is shown in the internal deformation characteristics of the honeycomb structure. In *X1*-H-RP in Figure 7a, a “C” shape folding interface occurs near the impact section as well at 1000 μs. The “C” interface and the upper interface form a lip-shaped interface. Then, an “X” interface also occurs near the impacted section and a “C” interface also occurs at the opposite from the blast at 1360 μs. At 1800 μs, the honeycomb structure is also rebounded because of the elastic properties. In the *X2*-H-RP in Figure 7b, the honeycomb sample is also expanded, but the honeycomb structure is not compacted. This expansion changes the shape of the cellular structure of the honeycomb from a convex hexagon to a quadrilateral as well at 890 μs. At 1260 μs, the impacted region also forms a “∇“ shape area, consisting of a series of horizontal lines. Besides, the deformation similarity is also verified in the cases of medium (20 g) and large explosives (40 g).

Different from the deformation characteristics of the RP-H-RP, a counter-intuitive deformation phenomenon appears near the explosive load surface in the H-RP. In this phenomenon, the shape of the loading wave is a concave face while the deformation surface of the honeycomb structure is a convex face in in-plane directions, as shown in Figure 7. It’s worth mentioning that the counter-intuitive phenomenon always exists, with the increase of equivalent explosive masses.

### 4.4. Discussion on Deformation Mechanism

Due to the low density of the honeycomb structure, the variation of the honeycomb structure deformation characteristics is essentially caused by the influence of the upper and lower panels on the propagation of the blast wave in these models. 

The propagation process of the shock wave in the honeycomb structure is shown in Figure 8a for RP-H-PR and Figure 8b for H-RP. In Figure 8, the shock wave just reaches the upper interface of the honeycomb at the moment T1. The wave is a concave interface. Then, the shock wave is reflected at the moment T2, whose shape is a convex interface because of the fixed bottom panel. T3 represents the moment when the reflected shock wave reaches the upper interface and reflects again. For the RP-H-RP with an upper panel, the concave shock wave is reflected again because of the reflection of the upper rigid panel. However, for the H-RP without the upper panel, the reflected wave is a concave sparse wave due to the upper interface, which is a free surface. Therefore, the “C”-shaped interface in the honeycomb is actually the propagation of the interface of the shock wave in the honeycomb structure. The shape of the ceiling surface of the honeycomb structure is the same as that of the convex shock wave, rather than the concave one. This is the reason for the counter-intuitive deformation phenomenon.

Two typical locations are selected from the honeycomb to analyze the honeycomb deformation distribution. Particle 1 is a particle in the middle of the upper interface of the honeycomb, indicated by a red circle. Particle 2 is a particle in the right area of the interface on the honeycomb, represented by a green circle, as shown in Figure 8. At the moment T3, the motion state of the two particles is shown in Figure 9. The particle velocity is the same as the wave direction in the shock wave, while the particle velocity is opposite in the sparse wave. In Figure 9, *v*1 is the velocity after the action of the upward convex shock wave, *v*2 is the velocity after the action of the reflected wave at the upper interface, and *v*0 is the velocity of the upper panel. The directions of these speeds is shown in Figure 9. For RP-H-RP in Figure 9a, the direction of *v*1 and *v*2 is opposite, so both particle 1 and particle 2 have moved downward at the velocity of *v*0, which is consistent with the deformation state in Figure 8. However, For H-RP in Figure 9b, the *v*1 and *v*2 are moving in the same direction, so Particle 1 is moving up at (*v*1 + *v*2), and Particle 2 is moving up at *v*4, which is the vertical component of (*v*1 + *v*2). Therefore, Particle 1 is above particle 2 because *v*4 is less than (*v*1 + *v*2). In addition, *v*3, the horizontal component of (*v*1 + *v*2), causes Particle 2 to shift in the horizontal direction, so the upper interface will expand to both sides.

## 5. Analysis of Energy and Stress Properties

### 5.1. Energy Distribution

For RP-H-RP, the total energy is divided into five parts at different moments, internal energy of SE, internal energy of DE, kinetic energy of rigid plate (marked as RP), kinetic energy of SE, and kinetic energy of DE, respectively. In order to compare the energy distribution of each part during the compression process, the energy ratio *ζ* is defined as follows,
(1)$$\zeta =\frac{E_{\mathrm{part}}}{E_{\mathrm{total}}}\times 100\%$$
where *E*_part_ represents one or more of the five components mentioned above, *E*_total_ is the total energy of the model.

Once the shock wave reaches the rigid plate in *X1*-RP-H-RP (Figure 10) or *X2*-RP-H-RP (Figure 11), most of the energy is converted immediately to the kinetic energy of the rigid plate (76.5% in overall response mode, 77.5% in transitional response mode, 59% in local response mode, 87.5% in expansion mode, and 62.0% in the local inner concave mode), and a little becomes the internal energy of honeycomb structure. Then, a partition of the kinetic energy of RP and the elastic energy of the honeycomb are transferred into the kinetic energy of the honeycomb. Before bouncing back of the rigid plate, the kinetic energy of all the parts decreases. However, the internal energy of the SE increases greatly and the internal energy of the DE increases slightly. With the increase of the explosive yield, the effect becomes more obvious. In addition, the kinetic energy of the SE and DE remains basically the same in the process.

According to Figure 10 and Figure 11, the deformation mode of the honeycomb can be judged from the kinetic energy and internal energy of DE. If the kinetic energy of DE is less than the internal energy of DE, the deformation mode of the honeycomb presents as an overall response or expansion mode. Otherwise, the deformation mode of the honeycomb is the local response mode.

For the H-RP, the total energy is divided into four parts, the kinetic energy and internal energy of SE and the kinetic energy and internal energy of DE, respectively, as illustrated in Figure 12. Once the shock wave reaches the ceiling of the honeycomb, 79.6% of the total energy is distributed to the kinetic energy of the DE for *X1*-H-RP and 76.3% of the total energy is distributed to kinetic energy of SE for *X2*-H-RP. In the subsequent process, the kinetic energies of SE and DE are basically equal. Finally, the total energy is converted to internal energy of SE.

It is noted that SE produces plastic deformation and converts total energy into deformation energy, while DE merely transfers energy in the form of kinetic energy. Therefore, in order to improve the energy absorption effect of a honeycomb structure, DE can be involved in the plastic deformation process through the optimization design. This provides a good way to enhance energy absorption.

Although the RP-H-RP and H-RP differ only by the ceiling rigid plate, the plate has an obvious effect on the energy absorption of the honeycomb structure. The internal energy of the SE in the -RP and RP-H-RP is compared in Figure 13. Once the shock wave reaches, 41.4% is absorbed for H-RP while 19.9% for RP-H-RP in the *X1* direction and 71.6% is absorbed for H-RP while 23.8% for RP-H-RP in the *X2* direction, respectively. Besides, the H-RP has more proportion of the energy, including elastic energy and plastic deformation energy of SE. However, the plate does not affect the energy absorption rate of the honeycomb structure.

### 5.2. Stress Analysis

For a buffering material, the peak stress is an important parameter to access the buffering ability of honeycombs. In order to describe the stress attenuation effect of the honeycomb structure, the buffering coefficient *η* is defined as follows,
(2)$$\eta =\frac{\sigma_{\mathrm{cp}}-\sigma_{\mathrm{bp}}}{\sigma_{\mathrm{cp}}}\times 100\%$$
where $\sigma_{\mathrm{cp}}$ is the initial contact peak stress between the honeycomb and the ceiling rigid plate and $\sigma_{\mathrm{bp}}$ is the initial contact peak force between honeycomb and the bottom rigid plate. The large buffering coefficient is beneficial to the structure buffering effect.

For 10 g RP-H-RP, as shown in Figure 14a and Figure 15a, the ceiling rigid plate gets the initial peak stress and about 15 μs later, the peak stress occurs in the bottom rigid plate, but its amplitude is decreased. Then, the stress in the ceiling rigid plate decreases rapidly and the stress in the bottom rigid plate increases. They are equal at about 800 μs. In the next stage, the stress response of the bottom plate is a platform. It is larger than the stress in the ceiling plate. It has a similar characteristic for 20 g RP-H-RP, as shown in Figure 14b. However, the bottom plate has a longer platform stress response. The stress in the bottom rigid plate increases rapidly at the final state for 40 g RP-H-RP in Figure 14c and Figure 15b. At this moment, the honeycomb specimen has been crushed. Besides, in different deformation modes, *η* is calculated as shown in Table 2. It can be seen that *η* is not correlated with the equivalent explosive masses, which is 73.06% in the *X1* direction and 65.72% in the *X2* direction, though a small rise with the masses. Therefore, the buffering effect in the *X1* direction is better than that in the *X2* direction.

## 6. Conclusions

Lightweight aluminum honeycomb is a nice bionic material to absorb energy under explosion. Two finite element models of honeycombs whose parent material is lightweight aluminum alloy 3003H18, including rigid plate-honeycomb-rigid plate (RP-H-RP) and honeycomb-rigid plate (H-RP) are created to investigate the in-plane deformation properties of honeycombs subjected to air blast. Conclusions can be drawn as follows,

(1) At the different equivalent explosive masses, there are three deformation modes in the *X1* direction, i.e., the overall response mode in the small equivalent explosive masses case in which there is a “C” or multi “C” folding shape, transitional response mode in the middle equivalent explosive masses case, in which there is an “I” and “C” folding shape, and the local response mode in the large equivalent explosive masses case in which there is only an “I” folding interface. There are mainly two deformation modes in the *X2* direction, expansion mode and local inner concave mode. In the expansion mode, the shape of the convex cells is changed into concave and some “I” shape folding interfaces form “∇“ shape area. In the local inner concave mode, “I” and “bow” interfaces appear successively, and finally the “M” interface is formed.

(2) The mechanism of deformation characteristics under different explosive equivalent conditions is given by wave propagation, and the counterintuitive phenomenon is explained due to the reflection of circle shock wave by the bottom plate but there is no secondary reflection on the ceiling plate in the H-RP.

(3) The honeycomb structure can absorb most of the energy carried by a shock wave. In the energy absorption process, it is mainly the deformation energy of the cell single edges with 0.04 mm thickness (marked as SE) that plays a major role. However, there is a low proportion of the internal energy of the cell double edges with 0.08 mm thickness (marked as DE). It provides a new way to improve the energy-absorbed capacity of honeycombs.

(4) The honeycomb structure can effectively attenuate the stress peak of a shock wave. Besides, the buffering effect in the *X1* direction is better than that in the *X2* direction.

The model presented is not yet validated experimentally; therefore, it represents a preliminary numerical approach of the deformation behavior of honeycomb structures under direct and indirect explosion.

## Figures and Tables

**Figure 1 materials-12-02222-f001:**
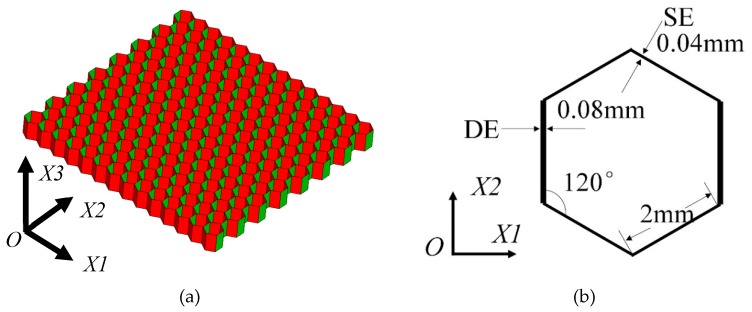
(**a**) A regular hexagonal honeycomb and (**b**) the sketch of a single unit cell.

**Figure 2 materials-12-02222-f002:**
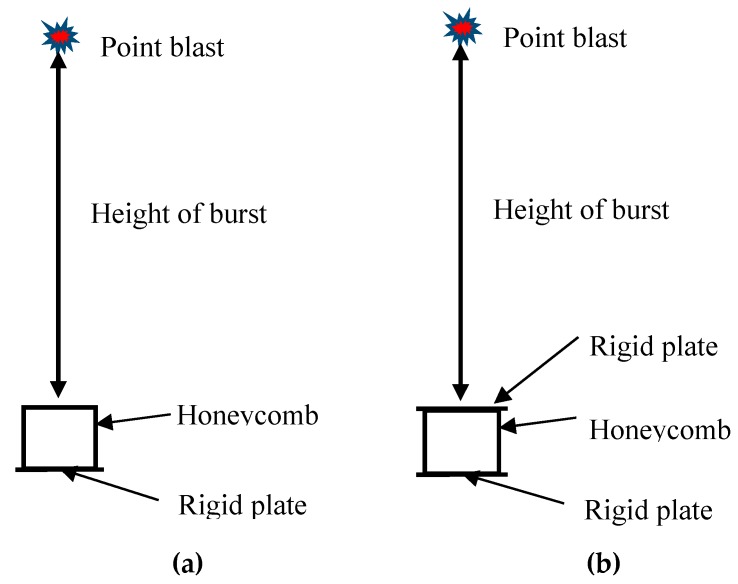
Two types of sketch models under air blast, (**a**) honeycomb-rigid plate and (**b**) rigid plate-honeycomb-rigid plate.

**Figure 3 materials-12-02222-f003:**
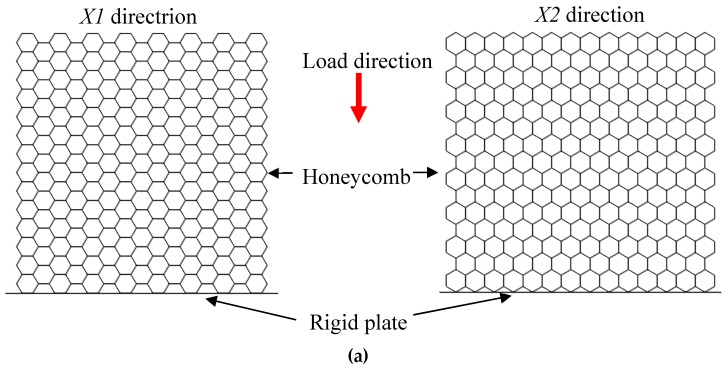
Equivalent finite element models, (**a**) honeycomb-rigid plate (H-RP) in the *X1* and *X2* direction; (**b**) rigid plate-honeycomb-rigid plate (RP-H-RP) in the *X1* and *X2* direction.

**Figure 4 materials-12-02222-f004:**
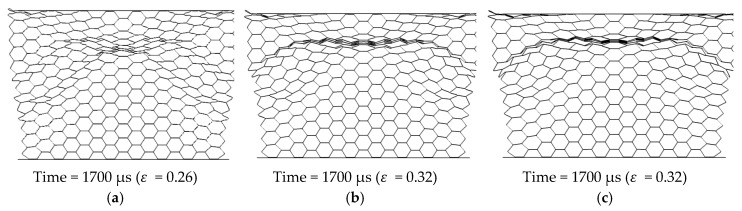
The deformation characteristics of three sizes of grids, (**a**) mesh size = 0.5 mm, (**b**) mesh size = 0.25 mm, and (**c**) mesh size = 0.125 mm.

**Figure 5 materials-12-02222-f005:**
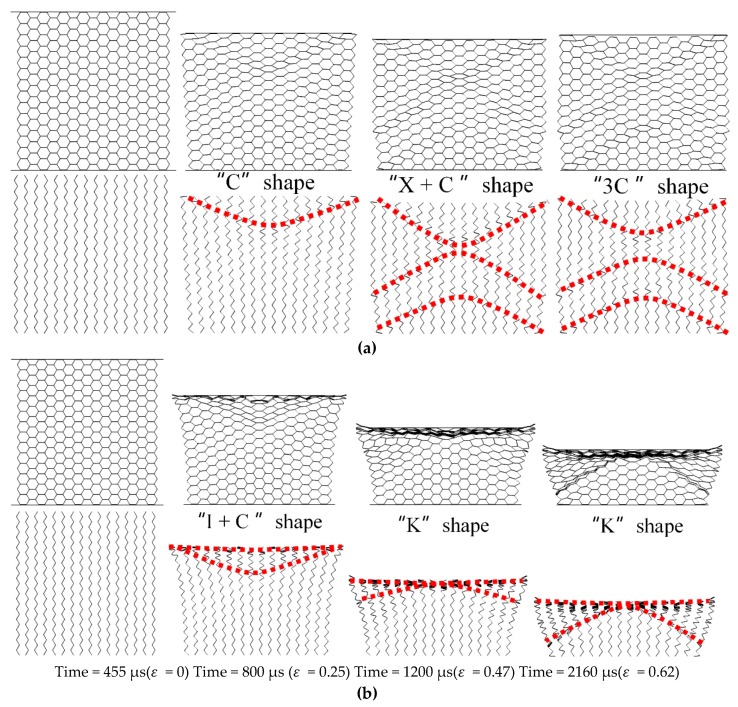
Deformation characteristics in the *X1*-RP-H-RP. (**a**) Overall response mode, (**b**) Transitional response mode, and (**c**) Local response mode.

**Figure 6 materials-12-02222-f006:**
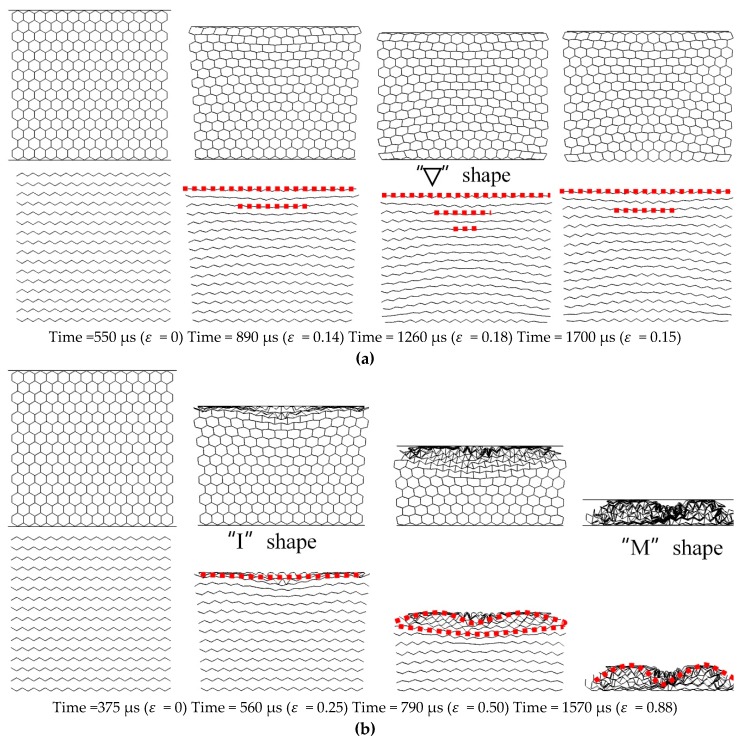
Deformation modes in the *X2*-RP-H-RP, (**a**) Expansion mode and (**b**) Local inner concave mode.

**Figure 7 materials-12-02222-f007:**
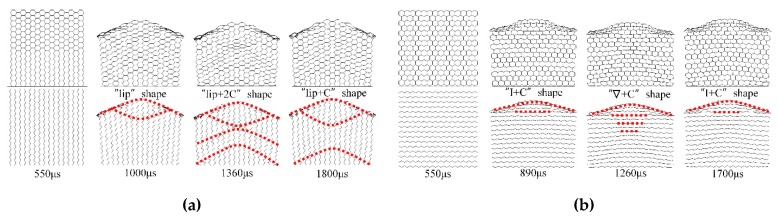
Deformation modes in the H-RP. (**a**) 10 g *X1*-H-RP and (**b**) 10 g *X2*-H-RP.

**Figure 8 materials-12-02222-f008:**
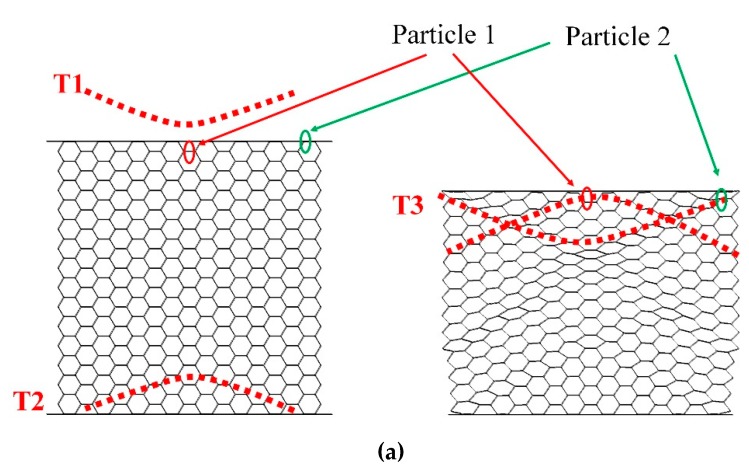
Wave propagation at different moments. (**a**) RP-H-RP and (**b**) H-RP.

**Figure 9 materials-12-02222-f009:**
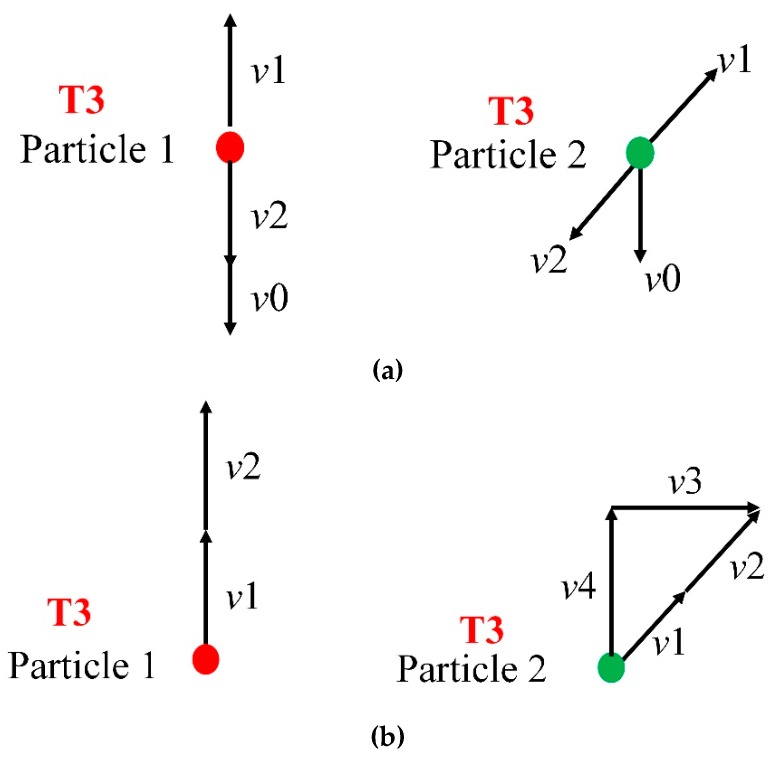
Schematic diagram of particles’ motion at the moment T3, (**a**) the particles near the top with a rigid plate and (**b**) the particles near the top without a rigid plate.

**Figure 10 materials-12-02222-f010:**
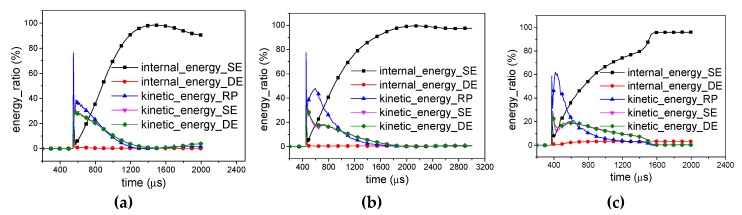
Energy distribution in *X1*-RP-H-RP. (**a**) Overall response mode; (**b**) Transitional response mode; (**c**) Local response mode.

**Figure 11 materials-12-02222-f011:**
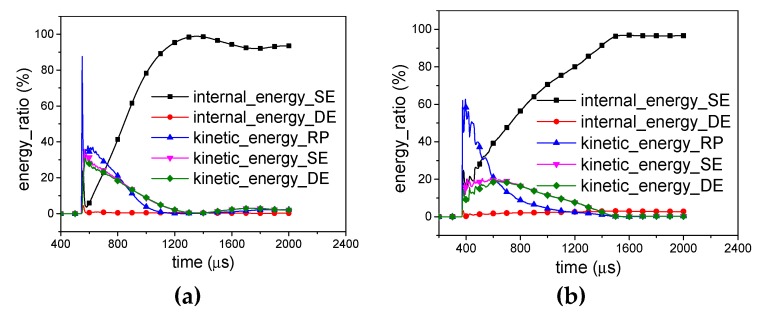
Energy distribution in *X2*-RP-H-RP. (**a**) Expansion mode and (**b**) Local inner concave mode.

**Figure 12 materials-12-02222-f012:**
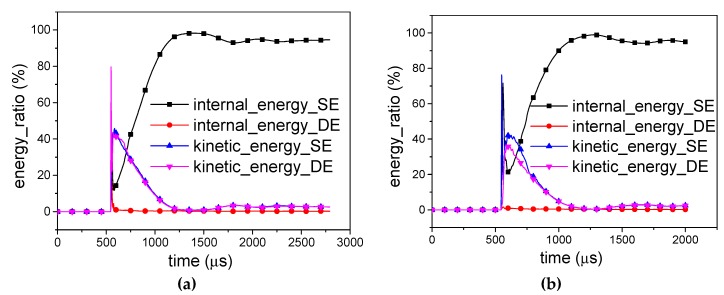
Energy distribution of H-RP. (**a**) *X1*-H-RP-10g and (**b**) *X2*-H-RP-10g.

**Figure 13 materials-12-02222-f013:**
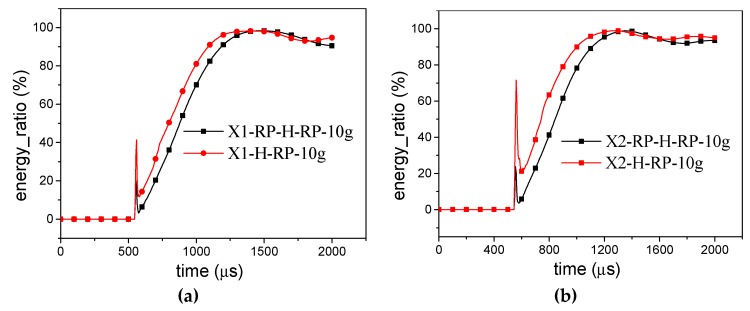
Comparison of energy distribution between RP-H-RP-10g and H-RP-10g (**a**) in the X1 direction and (**b**) in the X2 direction.

**Figure 14 materials-12-02222-f014:**
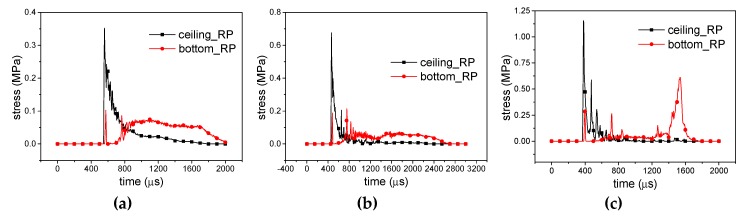
Stress-time curves of *X1*-RP-H-RP; (**a**) Overall response mode; (**b**) Transitional response mode and (**c**) Local response mode.

**Figure 15 materials-12-02222-f015:**
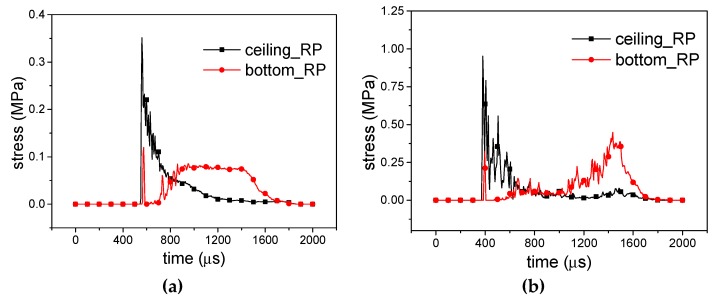
Stress-time curves of *X2*-RP-H-RP. (**a**) Expansion mode and (**b**) Local inner concave mode.

**Table 1 materials-12-02222-t001:** Material parameters.

Parts	Density/(g·cm^−3^)	Elastic Modulus/(GPa)	Poisson Ratio	Ultimate Strength/(MPa)
Honeycomb	2.7	69	0.33	289.6
Steel plate	7.83	210	0.3	-

**Table 2 materials-12-02222-t002:** The buffering effect under different equivalent explosive masses.

Cases	Deformation Modes	*σ_cp_*/MPa	*σ_bp_*/MPa	*η*/%
10 g *X1*-RP-H-RP	Overall	0.35	0.10	70.86
20 g *X1*-RP-H-RP	Transitional	0.68	0.19	72.24
40 g *X1*-RP-H-RP	Local	1.16	0.28	76.07
Average		-	-	73.06
10 g *X2*-RP-H-RP	Expansion	0.35	0.12	65.79
40 g *X2*-RP-H-RP	Local inner concave	0.95	0.32	66.08
Average		-	-	65.72

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
