# Peer review of "Numerical Simulation on In-plane Deformation Characteristics of Lightweight Aluminum Honeycomb under Direct and Indirect Explosion"

_materials, 2019, doi:10.3390/ma12142222_

Round 1

Reviewer 1 Report

The article has a serious flaw. The paper focuses only on numerical simulation only without any comparison to real experiment. In my opinion papers like that shouldn't be published at all and additional experiments are needed to prove the correctness of simulation. I recommend not to publish it.

Additional remarks:

- L124: Why plate of a 0.08 mm thickness is considered as rigid one? Why it is assumed it has only 1 D.O.F?

- Were the simulation done in 3D or 2D? It seems they were conducted in 2D which is oversimplification since the height of cells equal to 4mm only. L142 All nodes are restricted in the out-of-plane direction, but in reality, they would translate in the out-of-plane direction.

- If the DE and SE were connected by common nodes, and all the elements have thickness assigned (0.04 mm or 0.08mm) how authors dealt with the initial contact at joining junctions?

- L139: the description of the numerical model is too detailed. What is the benefit for readers from information about "cm-g-us". It is not important even for the group willing to recreate this simulation.

- L146: Why the scaling wasn't prevented during simulation? Since the scale factor for computed time step is 0.9 it wouldn't cost much more of computational time

- The naming of modes: like "C", "K", "I" is misleading. The drawing should be prepared to briefly present those modes.

- What was the temper of aluminum alloy used? Have the authors conducted any material tests?

- Authors often used deformation term like in L176. What do they mean by deformation? Is it the ratio of current deflection to initial height? If yes this is not "deformation". Additionally you the term "deformation" and "compressive strain" seems to be used interchangeably

Reviewer 2 Report

In this paper the deformation properties of honeycomb under dynamic impact or indirect explosive loading have been analyzed. The topic is interesting and is within the scope of Journal; I recommend publication of this paper after a revision in agreement with the following suggestions to the Authors:

1)      The Authors have not adequately analyzed the state of art about of the topic analyzed in this paper;

2)      The Authors should include in the paper the innovative contribution that this work leads to the actual state of the art;

3)      The Authors should highlight in a better way the goals obtained in their work;

4)      The Authors should pay more attention to the graphical resolution of the figures;

5)      The Authors must proof check their manuscript for few typos.  

The analysis of these aspects can greatly improve the paper.

Reviewer 3 Report

Numerical Simulation on In-plane Deformation 2 Characteristics of Lightweight Aluminum 3 Honeycomb under Direct and Indirect Explosion

Reviewer comments:

Honeycomb cells are used as core material in lightweight sandwich structures. Their usage is not limited to aerospace and aircraft applications. The main task of honeycomb core is to provide enough shear and compression strength to securely keep the distance between both outer sandwich cover sheets during normal operation. This greatly enhances the bending stiffness of the sandwich compared with solid material of same weight. If the strength is surpassed, the honeycomb structure collapses while an amount of energy is dissipated. Using metal (for example aluminum) as core material leads to a high energy dissipation due to plastic deformation. One possible application of those absorber materials is the usage as shield structure against blast load caused by an explosion.

The authors investigate the deformation characteristics of aluminum honeycomb material and the shockwave propagation in the material in this total numerically driven study. Three explosion energy levels were investigated with two different load boundary condition scenarios. Energy distributions and force progressions are compared for several load and blast energy cases.

With in-plane pressure, the study focuses on a very special load scenario for honeycomb structures. It is an interesting analysis of shock waves acting on an unshielded honeycomb cell boundary. However, the technical impact is not very large. The paper should be published with some changes.

Major issues:

(1) The in-plane loading of the honeycomb due to blast seems to be a very special case and the technical relevance is not really clear to me. Typically the honeycomb is embedded between sandwich cover sheets, which provide an additional support against wrinkling. Furthermore the free in-plane deformation (as modelled in the study) is not possible, because the core material is firmly tied to the outer cover sheets. This leads to a complete different loading scenario of such structures in reality (main blast loading direction is normal to the sandwich plane). The authors should provide at least one example of an application of such a structure. How is the free deformation realized while the blast loads only act in the in-plane directions (surrounding guidance structure or something else). Maybe there is an example in literature of the use of such a structural load case. Or maybe it is imaginable to use large blocks of honeycomb that have a large dimension in X3-direction (according to Fig.1(a)). The authors should address this point.

(2) There is some missing information regarding the model that has to be given. Especially the following:

-What is meant with “ultimate stress” (Table 1): elastic limit (yielding) or ultimate strength (damage)

-If yielding was meant with ultimate stress: what is the influence of damage mechanisms? Is the structure subjected to rupture in reality? Is this part of future research?

-Yielding and damage: typically both are dependent on the strain rate. A statement should be given how important this strain rate dependency is in real structures/materials.

-In general, it is dangerous to assume that a model behaves right without an experimental validation. The authors should compare the results with experiments in literature. Or, if this is not possible, the authors have to state that the model is not validated and the model is only used as preliminary numerical example and not to predict real honeycomb structure behavior.

-Was there a strain hardening model in the analysis? What kind? Which parameters? Strain hardening is an unneglectable effect.

-Which strain is meant in Fig.4+5 + in text: elastoplastic or purely plastic; purely compressive strain?

-In line 112 it was stated that the mesh density is important and in line 153 a conducted mesh density analysis was mentioned. The authors should give the results of this analysis or otherwise explain why the element size was set to 0.125mm.

-The comparability is not really given in Fig. 4 + 5 for the reader; times and strains are completely different. Are the time output steps chosen according to (time at final position)/3 ? If this is the case the authors should state this.

-The initial velocities due to blast load with (10/20/40)g TNT would be useful in Fig. 4+5. At least in the text.

-Is hourglassing a problem? The authors use an under-integrated type2 shell element. Hourglassing often appears with this shell type.

(3a) “Counter-intuitive phenomenon” in line 238 and abstract+conclusion: I am really not sure, whether it is caused by the shock wave in the material and wave reflection at the lower rigid plate. Isn’t it possible that the convex shape is caused by the missing shield (upper rigid plate), because the shape occurs at a very early stage in the simulation? In other words: is it possible that the free upper honeycomb edges were simply blown away by blast pressure?

(3b) lines 178-180: Is it possible that the deformation of those cells occurs between the plot-states? If I estimate the time a shockwave travels through a block of solid aluminum of corresponding dimensions, I get: t_travel=s_block/v_speed-of-sound=48mm/(3000000mm/s)=1.6E-05s=16µs à This is on the level of the plot-state output interval of 5µs (line 146).

Minor issues:

-line 10: energy-absorbing

-l. 33: “orthotropic” always means 3 special orthogonal material directions

-l. 42-45: This sentence is hardly understandable. Reformulation and maybe division into smaller sentences could improve readability.

-l. 57: “it is unavoidable that honeycomb is subjected to explosive loading”: Maybe in a war region… But in Civil Engineering explosions are very unusual…

-l. 94: why is this dimension of 47.6x46.0mm² chosen?

-l. 112: “grid density effect”: maybe “discretization of the structure” is better?

-l. 121-122: “DE” and “SE” are not marked in Fig.1(b)

-l. 125: delete second “be” in “can be only be translated”—and maybe it’s better to explain this with degree of freedom of the rigid body?

-l. 126-127: “Both models are studied…to study the effect”

-l. 132: “…of TNT can be gotten in Ref…” à “of TNT were taken from Ref…”

-l. 140: all elements are 0.04mm thick? Are the 0.08mm double edges composed from 2 0.04mm edges? Are they tied together via tied-contact?

-l. 136: “cannot be set to infinity…to ensure stability of calculation”: The rigid body algorithm stands for infinite stiffness. Density and stiffness defined in *MAT_RIGID are only used to compute the contact stiffness. This has no influence on stability.

-l. 142: “…restricted in the out-of-plane direction”: maybe it is also better to explain this with nodal degree of freedom

-l. 145: better: “…to provide simulation accuracy”

-l. 152: K-file

-l. 155: “If” not with capital letter

-l. 180: “payload” better: only “load” or “deformation”

-l. 195: compression strain

-l. 195: delete “to” in “The sample reaches to the compaction state…” à also in l. 261, 276

-l. 202 and Fig.5: The reader expects also the 20g-TNT model here. Please add a sentence that 20g is similar to 10g (or 40g if this is the case)

l. 228: “departure impacted field”? Maybe “opposite from the blast” or “at lower rigid plate” or something like that

-Section 4.4: This is not conclusively derieved for me. See also (3a) above. Please add more content on this or at least an explaining picture.

l. 259: E_part and E_total are above the line.

l. 294: At this point it is not clear where the stresses come from. The rigid plates have no stress. I guess the authors used the contact forces together with the rigid plate area to calculate the stress. Please add an explanation in the text.

l. 295: The sentence “The larger…is, the better…is.” has to be improved.

l. 298: “…decreases a lot” is not very precise.

l. 305: “… eta has nothing to do with …exploding masses” à I see a small but clear rise with masses.

l. 338: First reference entry has to be indented.

Round 2

Reviewer 1 Report

Please conduct the FEM simulation in which the plate is not assumed as rigid. If the results will be the same than we can say that this simplification is acceptable. Isn't it right that even if the standalone plate is tested there should be bending of the plate?

Point 2: L124: Why plate of a 0.08 mm thickness is considered as rigid one? Why it is assumed it has only 1 D.O.F?

Response 2:

(1) Plate of a 0.08 mm thickness is considered as rigid one in order to compare the differences of deformation and energy distribution between direct explosion load and indirect explosion load on honeycomb structures. Because the strength of the honeycombs is very low compared with the strength of the upper plate of a 0.08 mm thickness. The deformation of the upper plate can be ignored under distant explosive load. Therefore, the plate of a 0.08 mm thickness is set as a rigid body to conduct the work.

(2) The plate of a 0.08 mm thickness has only 1 D.O.F, because it is a 2D problem to study the in-plane deformation of honeycomb structures. Especially, the explosive loading on the honeycomb structure is axisymmetric. Therefore, the rotation D.O.F. in the direction of loading and the rotational and translated freedom in the out-of-plane direction can be ignored.

I partially get the authors approach. The problem once again is that you conduct a purely theoretical test with no reference to a real example. In real life scenarios, a global bending would be seen. To guarantee there is no out-of-plane deformation additional elements preventing this behaviour should be installed in the real model.

Point 3: Were the simulation done in 3D or 2D? It seems they were conducted in 2D which is oversimplification since the height of cells equal to 4mm only. L142 All nodes are restricted in the out-of-plane direction, but in reality, they would translate in the out-of-plane direction.

Response 3:

The simulation were done in 3D to explore the two-dimensional deformation properties of honeycombs. According to the existed analogous literature [1,2], it is feasible to fix the thickness direction to 4 mm for numerical simulation. The focus of the work is the in-plane mechanical properties, rather than in the out-of-plane. The height of cells equals to 2 mm in Ref [1]. In their work, the honeycomb structure is impacted by a rigid flat plane with constant velocity. The same height is set in Ruan’ work [2]. They have been validated effectively.

Honeycomb is a type of cellular material with a two-dimensional array of hexagonal cells. The deformation in out-of-plane direction is the same. In reality, this honeycomb structure is applied to the distant explosive wave. In this case, the size of the honeycomb structure in the out-of-plane (4 mm) compared to the radius of the blast wave is very small, so all nodes can be seen restricted in the out-of-plane direction.

What I was trying to say is that you only reduced computation time by about 10%. So, I recommend you to disable scaling (scaling factor 1). It is not considered as a mistake by I see no benefit of scaling.

Point 6: L124: Why the scaling wasn't prevented during simulation? Since the scale factor for computed time step is 0.9 it wouldn't cost much more of computational time.

Response 6:

I agree with you. It is possible to scale the finite element model to conduct the work, which can decrease the time cost in the numerical simulation for the cellular structures. However, Wang [1] has been verified that when the total size in ou-of-plane direction is approximate to the cell length, the scaling method is limited. We are not sure that it is suitable to set the scaing, which can keep the same regularity to the origin model. We are intended to conduct the scaling work in the future, which may be very interesting.

Reference

[1] Z. Wang, S. Yao. An equivalent method to cell magnification of aluminum honeycomb under out-of-plane compression. Explosion and Shock Waves, 2013, 33(3):269-274.

This explanation seems satisfactory, but it raised another question. Was the production process modeled as well? The honeycomb is strain hardened along bending edges and the properties along edges differ from initial properties of the material. This may lead to substantial differences in deformation nodes when comparing models including these changes and model not including these changes. 

Point 8: What was the temper of aluminum alloy used? Have the authors conducted any material tests?

Response 8:

Commercially hexagonal honeycombs core with unit cell length (l) of 2 mm, manufactured by a stretch forming method are used in this study, as shown in Fig.1. The honeycombs are produced by the HANGYU, a honeycomb company in Hubei province in China. Therefore, the tempering process of aluminum alloy was not recorded by the authors. However, mechanical properties of honeycomb’s parent material (AL3003H18) are investigated by two types of material tests, quasi-static and dynamic tensile loading, respectively.

Based on the standard GB/T 228.1-2010, quasi-static experiments are designed and conducted with 2 mm/min. The thickness of all expemental specimens is 2.16 mm. Results of three repeated experiments are in consistency perfectly. Dynamic tensile specimens are designed and the effective size is 4mm×4mm×2.16 mm. Four round corners near the field are designed to help stress balance better during dynamic tension process. Dynamic tension test are conducted using split Hopkionson tension bar (SHTB). The specimens size of both cases are shown as follows,

Stress-strain curves of AL3003H18 are shown as follows, in which (a) is at quasi-static loading condition and (b) is at dynamic loading condition. In both cases, the entire process in the curves can be divided into three stages before failure of specimen. Part I is the elastic stage. Then status of specimen come into plastic stage during Part II. Part III occurs the necking phenomenon. In general, it is difficult to reach Part III for honeycomb materials due to the structural buckling.

It indicates that the ultimate strength at dynamic load is about 19.7% higher than  that at quasi-static  load, 289.6 MPa and 241.9 MPa respectively, which means that AL3003H18 is strain-rate sensitive material. Therefore, the dynamic ultimate strength at dynamic conditions is applied in the work of finite element analysis. Due to the limitation of the length of the paper, these material experiments are not placed in the new manuscript.

The points not mentioned here are explained thoughtfully.

Because of lack of validation of FEM model with the real experiment I regret to recommend this work not to be published. The explanation that there are other papers concerning only FEM modeling is not satisfactory.

Reviewer 2 Report

The paper is improved by the authors. I think that the article is interesting and deserves publication.
